# An Analysis of Composite Neural Network Performance from Function Composition Perspective

## Abstract

This work investigates the performance of a composite neural network, which is composed of pre-trained neural network models and non-instantiated neural network models, connected to form a rooted directed graph. A pre-trained neural network model is generally a well trained neural network model targeted for a specific function. The advantages of adopting such a pre-trained model in a composite neural network are two folds. One is to benefit from other's intelligence and diligence, and the other is saving the efforts in data preparation and resources and time in training. However, the overall performance of composite neural network is still not clear. In this work, we prove that a composite neural network, with high probability, performs better than any of its pre-trained components under certain assumptions. In addition, if an extra pre-trained component is added to a composite network, with high probability the overall performance will be improved. In the empirical evaluations, distinctively different applications support the above findings.

## 1 Introduction

Deep learning has been a great success in dealing with natural signals, e.g., images and voices, as well as artifact signals, e.g., nature language, while it is still in the early stage in handling sophisticated social and natural applications shaped by very diverse factors (e.g., stock market prediction), or resulted from complicated processes (e.g., pollution level prediction). One of distinctive features of the complicated applications is their applicable data sources are boundless. Consequently, their solutions need frequent revisions. Although neural networks can approximate arbitrary functions as close as possible (Hornik, 1991), the major reason for not existing such competent neural networks for those complicated applications is their problems are hardly fully understood and their applicable data sources cannot be identified all at once. By far the best practice is the developers pick a seemly neural network with available data to hope for the best. The apparent drawbacks, besides the performance, are the lack of flexibility in new data source emergence, better problem decomposition, and the opportunity of employing proven efforts from others. On the other hand, some adopts a composition of several neural network models, based on function composition using domain knowledge.

An emerging trend of deep learning solution development is to employ well crafted pre-trained neural networks (i.e., neural network models with instantiated weights), especially used as a component in a composited neural network model. Most popular pre-trained neural network models are well fine tuned with adequate training data, and made available to the public, either free or as a commercial product. During the training phase of composite neural network, the weights of pre-trained models are frozen to maintain its good quality and save the training time, while the weights of their outgoing edges are trainable. In some cases as in the transfer learning, the weights of pre-trained neural network are used as initial values in the training phase of composite neural network. It is intuitive that a composite neural network should perform better than any of its components. The ensemble learning (Freund & Schapire, 1997; Zhou, 2012) and the transfer learning (Galanti et al., 2016) have great success and are popular when pre-trained models are considered. However, the following example shows some aspects missed by these two methods, and requests for more complicated composite function.

**Example 1.** Assume there is a set of locations indexed as $X = \{(0,0),(0,1),(1,0),(1,0)\}$ with the corresponding values $Y = (0,1,1,0)$. Obviously, the observed function is the XOR (Goodfellow et al., 2016). Now consider three models: $f_1(x_1,x_2) := x_1$, $f_2(x_1,x_2) := x_2$, and $f_3(x_1,x_2) := x_1 x_2$. Their corresponding output vectors are $(0,0,1,1)$, $(0,1,0,1)$, $(0,0,0,1)$ with bit-wise accuracy 50%, 50%, 25%, respectively. This means that the AdaBoosting algorithm will exclude $f_1$ and $f_2$ in the ensemble since their coefficients are $\frac{1}{2}\ln\frac{1-50\%}{50\%} = 0$. On the other hand, in the transfer learning, $f_3$ is fine-tuned by applying the gradient descent method with respect to L2 loss on $wf_3 = wx_1 x_2$ to transfer the source task distribution to that of the target task. The result comes to $w = 0$, and $f_3$ is excluded. Now consider $g_1(x_1,x_2) = \alpha_1 f_1 + \alpha_2 f_2$ and apply the back-propagation method with respect to the L2 loss. The results are $\alpha_1 = \alpha_2 = \frac{1}{3}$, with loss $\frac{4}{3}$. If further define $g_2(x_1,x_2) = w_1 g_1 + w_2 f_3$, the back-propagation yields $g_2 = 3g_1 - 2f_3 = x_1 + x_2 - 2x_1 x_2$ with the output $(0,1,1,0)$. The final $g_2$ computes $Y$ with loss 0. This example shows the power of composite function.

**Composite Neural Network.** In the transfer learning, how to overcome the negative transfer (a phenomenon of a pre-trained model has negative impact on the target task) is an important issue (Seah et al., 2013). In the ensemble learning, it is well known that the adding more pre-trained models, it is not always true to have the better accuracy of the ensemble (Zhou et al., 2002). Furthermore, Opitz & Maclin (1999) pointed that the ensemble by boosting having less accuracy than a single pre-trained model often happens for neural networks. In the unsupervised learning context, some experimental research concludes that although layer-wise pre-training can be significantly helpful, on average it is slightly harmful (Goodfellow et al., 2016). These empirical evidences suggest that in spite of the success of the ensemble learning and the transfer learning, the conditions that composite neural network can perform better is unclear, especially in the deep neural networks training process. The topology of a composite neural network can be represented as a rooted directed graph. For instance, an ensemble learning can be represented as 1-level graph, while a composite neural network with several pre-trained models that each is designed to solve a certain problem corresponds to a more complicated graph. It is desired to discover a mathematical theory, in addition to employing domain knowledge, to construct a composite neural network with guaranteed overall performance. In this work, we investigate the mathematical theory to ensure the overall performance of a composite neural network is better than any a pre-trained component, regardless the way of composition, to allow deep learning application developer great freedom in constructing a high performance composite neural network.

**Contributions.** In this work, we proved that a composite neural network with high probability performs better than any of its pre-trained components under certain assumptions. In addition, if extra pre-trained component is added into a composite network, with high probability the overall performance will be improved. In the empirical evaluations, distinctively different applications support the above findings.

## 2 PRELIMINARIES

In this Section, we introduce some notations and definitions about composite neural network. Parameters $N$, $K$, $d$, $d_j$, $d_{j_1}$, and $d_{j_2}$ are positive integers. Denote $\{1,...,K\}$ as $[K]$ and $[K] \cup \{0\}$ as $[K]^+$. Let $\sigma : \mathbb{R} \to \mathbb{R}$ be a differentiable activation function, such as the Logistic function $\sigma(z) = 1/(1+e^{-z})$ and the hyperbolic tangent $\sigma(z) = (e^z - e^{-z})/(e^z + e^{-z})$. For simplicity of notation, we sometimes abuse $\sigma$ as a vector value function. A typical one hidden layer neural network can be formally presented as $w_{1,1}\sigma\left(\sum_{i=1}^d w_{0,i}\mathbf{x}_i + w_{0,0}\right) + w_{1,0}$. We abbreviate it as $f_{\sigma,\mathbf{W}}(\mathbf{x})$, where $\mathbf{W}$ is the matrix defined by $w_{1,1}, w_{1,0}, ..., w_{0,1}, w_{0,0}$. Recursively applying this representation can obtain the neural network with more hidden layers. If there is no ambiguity on the activation function, then it can be skipped as $f_{\mathbf{W}}(\mathbf{x})$. Now assume a set of neural networks $\{f_{\mathbf{W}_j}(\mathbf{x}_j)\}_{j=1}^K$ is given, where $\mathbf{W}_j$ is the real number matrix defining the neural network $f_{\mathbf{W}_j} : \mathbb{R}^{d_{j_1} \times d_{j_2}} \to \mathbb{R}^{d_j}$, and $\mathbf{x}_j \in \mathbb{R}^{d_{j_1} \times d_{j_2}}$ is the input matrix of the $j$th neural network. For different $f_{\mathbf{W}_j}$, the corresponding $d_j$, $d_{j_1}$ and $d_{j_2}$ can be different. For each $j \in [K]$, let $D_j = \{(\mathbf{x}_j^{(i)}, \mathbf{y}_j^{(i)}) \in \mathbb{R}^{(d_{j_1} \times d_{j_2}) \times d_j}\}_{i=1}^N$ be a set of labeled data (for the $j$th neural network). For each $i \in [N]$, let $\mathbf{x}^{(i)} = (\mathbf{x}_1^{(i)}, \ldots, \mathbf{x}_K^{(i)})$, $\mathbf{y}^{(i)} = (\mathbf{y}_1^{(i)}, \ldots, \mathbf{y}_K^{(i)})$, and $D = \{(\mathbf{x}^{(i)}, \mathbf{y}^{(i)})\}_{i=1}^N$.

For a **pre-trained** model (component), we mean $\mathbf{W}_j$ is fixed after its training process, and then we denote $f_{\mathbf{W}_j}$ as $f_j$ for simplicity. On the other hand, a component $f_{\mathbf{W}_j}$ is **non-instantiated** means $\mathbf{W}_j$ is still free. A deep feedforward neural network is a hierarchical acyclic graph, i.e. a directed tree. In this viewpoint, a feedforward neural network can be presented as a series of function compositions. For given $\{f_{\mathbf{W}_j}(\mathbf{x}_j)\}_{j=1}^K$, we assume $\theta_j \in \mathbb{R}^{d_j}$, $j \in [K]$, which make the product $\theta_j f_{\mathbf{W}_j}(\mathbf{x}_j)$ is well-defined. Denote $f_0$ as the constant function 1, then the liner combination with a bias is defined as as $\Theta(f_1, ..., f_K) = \sum_{j \in [K]^+} \theta_j f_j(\mathbf{x}_j)$. Hence, an $L$ layers of neural network can be denoted as $\Theta_{(L)} \circ \sigma \circ \cdots \circ \Theta_{(0)}(\mathbf{x})$. A composite neural network defined by components $f_{\mathbf{W}_j}(\mathbf{x}_j)$ can be designed as an directed tree. For instance, a composite neural network $\sigma_2\left(\theta_{1,0} + \theta_{1,1}f_4(\mathbf{x}_4) + \theta_{1,2}\sigma_1(\theta_{0,0} + \theta_{0,1}f_1(\mathbf{x}_1) + \theta_{0,2}f_{\mathbf{W}_2}(\mathbf{x}_2) + \theta_{0,3}f_3(\mathbf{x}_3))\right)$ can be denoted as $\sigma_2 \circ \Theta_1\left(f_4, \sigma_1 \circ \Theta_0(f_1, f_{\mathbf{W}_2}, f_3)\right)$, where $f_1$ and $f_3$ are pre-trained and $f_{\mathbf{W}_2}$ is non-instantiated. Note that in this work $D_j$ is the default training data of component $f_j$ of composite neural network, but $D_j$ can be different from the training data deciding the frozen weights in the pre-trained $f_j$.

Let $\langle \vec{a}, \vec{b} \rangle$ be the standard inner product of $\vec{a}$ and $\vec{b}$, and $|| \cdot ||$ be the corresponding norm. For a composite neural network, the training algorithm is the gradient descent back-propagation algorithm and the loss function is the $L_2$-norm of the difference vector. In particular, for a composite neural network $g_{\vec{\theta}}$ the total loss on the data set $D$ is

$$L_{\vec{\theta}}\left(\mathbf{x}; g_{\vec{\theta}}\right) = \langle \vec{g}_{\vec{\theta}}(\mathbf{x}) - \vec{y}, \vec{g}_{\vec{\theta}}(\mathbf{x}) - \vec{y} \rangle = ||\vec{g}_{\vec{\theta}}(\mathbf{x}) - \vec{y}||^2 \tag{1}$$

This in fact is $\sum_{i=1}^N \left(g(\mathbf{x}^{(i)}) - \mathbf{y}^{(i)}\right)^2$. By the definition of $g_{\vec{\theta}}(\cdot)$, this total loss in fact depends on the given data $\mathbf{x}$, the components defined by $\{\Theta_j\}_{j=1}^K$, the output activation $\sigma$, and the weight vector $\vec{w}$. Similarly, let $L(f_j(\mathbf{x}_j))$ be the loss function of a single component $f_i$. Our goal is to find a feasible $\vec{\theta}$ s.t. $L_{\vec{\theta}}(\mathbf{x}; g) < \min_{j \in [K]} L(f_j(\mathbf{x}_j))$.

## 3  PROBLEM SETTINGS AND RESULTS OVERVIEW

The problems considered in this work are as follows:

P1. What are the conditions that the pre-trained components must satisfy so that them can strictly improve the accuracy of the whole composition?

P2. Will more pre-trained components improve the accuracy of the whole composition?

Let $\vec{f}_j$ be the output vector of the $j$th pre-trained component, and $\mathcal{B}_K$ be the set of unit vectors in $\mathbb{R}^K$.

A1. Linearly Independent components (LIC) Assumption:
$\forall t \in [K], \nexists\{\beta_j\} \subset \mathbb{R}$, s.t. $\vec{f}_t = \sum_{j \in [K] \setminus \{t\}} \beta_j \vec{f}_j$.

A2. No Perfect component (NPC) Assumption:
$\min_{j \in [K]} \left\{ \sum_{i \in [N]} f_j(\mathbf{x}_j^{(i)}) - \mathbf{y}^{(i)} \right\} > \epsilon^*$, where $\epsilon^* > 0$ is constant.

Our results are as follows:

**Theorem 1.** *Assume the set of components $\{f_j(\mathbf{x}_j)\}_{j=1}^K$ satisfies LIC. Let $g$ be $\Theta(f_1, ..., f_K)$. With probability at least $1 - \frac{K}{\pi e^N}$, there is a vector $\vec{\theta} \in \mathbb{R}^K \setminus \mathcal{B}_K$ s.t. $L_{\vec{\theta}}(\mathbf{x}; g) < \min_{j \in [K]}\{L(f_j(\mathbf{x}_j))\}$.*

**Theorem 2.** *Assume the set of pre-trained components $\{f_j(\mathbf{x}_j)\}_{j=1}^K$ satisfies both NPC and LIC, and $g$ be $\sigma \circ \Theta(f_1, ..., f_K)$. Then with probability at least $1 - \frac{K}{\pi e^N}$ there exists $\vec{w}$ s.t. $L_{\vec{w}}(\mathbf{x}; g) < \min_{j \in [K]} L(f_j(\mathbf{x}_j))$.*

**Theorem 3.** *Assume the set of components $\{f_j(\mathbf{x}_j)\}_{j=1}^K$ satisfies LIC. Let $g_{K-1} = \Theta(f_1, ...f_{K-1})$ and $g_K = \Theta(f_1, ...f_K)$. With probability at least $1 - \frac{K}{\pi e^N}$, there is a vector $\vec{w} \in \mathbb{R}^K \setminus \mathbb{B}_{\mathbb{R}^K}$ s.t. $L_{\vec{w}}(\mathbf{x}; g_K) < L_{\vec{w}}(\mathbf{x}; g_{K-1})$.*

Theorem 1, and 2 together answer Problem P1, and Theorem 3 answers Problem P2.

## 4 RELATED WORK

Our framework is related but not the same with the models such as transfer learning. (Erhan et al., 2010; Kandaswamy et al., 2014; Yao & Doretto, 2010) and ensemble leaning (Zhou, 2012).

**Transfer Learning.** Typically transfer learning deals with two data sets with different distributions, source and target domains. A neural network, such as an auto-encoder, is trained with source domain data and corresponding task, and then part of its weights are taken out and plugged into other neural network, which will be trained with target domain data and task. The transplanted weights can be kept fixed during the consequent steps or trainable for the fine-tune purpose (Erhan et al., 2010). For multi-source transfer, algorithms of boosting based are studied in the paper (Yao & Doretto, 2010). Kandaswamy et al. (Kandaswamy et al., 2014) proposed a method of cascading several pre-trained layers to improve the performance. Transfer learning is considered as a special case of composite neural network that the transfered knowledge can be viewed as a pre-trained component.

**Ensemble (Bagging and Boosting).** Since the Bagging needs to group data by sampling and the Boosting needs to tune the probability of data (Zhou et al., 2002), these frameworks are different from composite neural network. However, there are fine research results revealing many properties for accuracy improvement (Džeroski & Ženko, 2004; Gashler et al., 2008; Zhou et al., 2002). For example, it is known that in the ensemble framework, low diversity between members can be harmful to the accuracy of their ensemble (Džeroski & Ženko, 2004; Gashler et al., 2008). In this work, we consider neural network training, but not data processing.

**Ensemble (Stacking).** Among the ensemble methods, the stacking is closely related to our framework. The idea of stacked generalization (Wolpert, 1992), in Wolpert's terminology, is to combine two levels of generalizers. The original data are taken by several level 0 generalizers, then their outputs are concatenated as an input vector to the level 1 generalizer. According to the empirical study of Ting and Witten (Ting & Witten, 1999), the probability of the outputs of level 0, instead of their values, is critical to accuracy. Besides, multi-linear regression is the best level 1 generalizer, and non-negative weights restriction is necessary for regression problem while not for classification problem. In (Breiman, 1996), it restricts non-negative combination weights to prevent from poor generalization error and concludes the restriction of the sum of weights equals to 1 is not necessary (Breiman, 1996). In (Hashem, 1997), Hashem showed that linear dependence of components could be, but not always, harmful to ensemble accuracy, while in our work, it allows a mix of pre-defined and undefined components as well as negative weights to provide flexibility in solution design.

**Recently Proposed Frameworks.** In You et al. (2017), Shan You et al. proposed a student-teacher framework where the outputs of pre-trained teachers are averaged as the knowledge for the student network. A test time combination of multiple trained predictors was proposed by Kim, Tompkin, and Richardt In Kim et al. (2017), that the combination weights are decided during test time. In above frameworks, the usage of pre-trained neural networks generally improves the accuracy of their combination.

## 5 THEORETICAL ANALYSIS

This section provides analyses of the loss function of composite neural network with the introduction of pre-trained components. For the complete proofs, please refer to Supplementary Material. Observe that for given pre-trained components $\{f_j\}_{j=1}^K$, a composite neural network can be defined recursively by postorder subtrees search. For instance, $\sigma_2 \circ \Theta_1(f_4, \sigma_1 \circ \Theta_0(f_1, f_2, f_3))$ can be presented as $\sigma_2 \circ \Theta_1(f_4, g_1)$, and $g_1 = \sigma_1 \circ \Theta_0(f_1, f_2, f_3)$. Without loss of generality, we assume $d_j = d = 1$ for all $j \in [K]$ in the following proofs. We denote $\vec{f_j}$ the vector $(f_j(\mathbf{x}^{(1)}), \cdots, f_j(\mathbf{x}^{(N)}))$, as the sequence of $f_j$ during the training phase. Similarly, $\vec{y} := (y^{(1)}, \cdots, y^{(N)})$. Let $\vec{e_j}$ be an unit vector in the standard basis of $\mathbb{R}^K$ for each $j \in [K]$, i.e. $\vec{e_1} = (1, 0, \cdots, 0)$ and $\vec{e_2} = (0, 1, 0, \cdots, 0)$, etc. Let $\mathcal{B}_K$ be the set containing all these standard unit-length basis of $\mathbb{R}^K$.

**Theorem 1.** *Assume the set of components $\{f_j(\mathbf{x}_j)\}_{j=1}^K$ satisfies LIC. Let $g$ be $\Theta(f_1, ..., f_K)$. With probability at least $1 - \frac{K}{\pi e^N}$, there is a vector $\vec{\theta} \in \mathbb{R}^K \setminus \mathcal{B}_K$ s.t. $L_{\vec{\theta}}(\mathbf{x}; g) < \min_{j \in [K]}\{L(f_j(\mathbf{x}_j))\}$.*

*Proof.* (**Proof Sketch**) The whole proof is split to Lemma 5.1,5.2,5.3. Note that $g(\cdot)$ is the linear combination of $\vec{\theta}$ and $\{f_j(\mathbf{x}_j)\}_{j=1}^K$. It is well known (Friedman et al., 2001) that to search the minimizer $\vec{\theta}$ for $L_{\vec{\theta}}$, i.e. to solve a least square error problem, is equivalent to find an inverse matrix defined by $\{f_j(\mathbf{x}_j)\}_{j=1}^K$. Since $\{f_j(\mathbf{x}_j)\}_{j=1}^K$ satisfy LIC, the inverse matrix can be written down concretely, which proves the existence. Furthermore, if this solved minimizer $\vec{\theta}^*$ is not $\vec{e}_s$ for some $s \in [K]$ then the $g_{\vec{\theta}*}$ has lower loss than $f_s$. Lemma 5.3 argues that the probability of $\vec{\theta}^* = \vec{e}_s$ is at most the probability of the event $\langle \vec{f} - \vec{y}, \vec{f} \rangle = 0$, where $\vec{f}$ is uniformly taken from the vector set of the same length of $\vec{f} - \vec{y}$. $\square$

The statements of Lemmas needed by previous Theorem are as follows.

**Lemma 5.1.** *There exists $\vec{\theta} \in \mathbb{R}^{K+1}$ s.t. $L_{\vec{\theta}}\big(\mathbf{x}; \Theta_{(0)}(f_1, ... f_K)\big) \le \min_{j\in[K]^+}\{L(f_j(\mathbf{x}_j))\}$.*

This Lemma deals with the existence of the solution of the inequality. But our goal is to find a solution such that the loss is strictly less than any pre-trained component.

**Lemma 5.2.** *Denote $\mathbb{I}_{L_{\vec{\theta}}}$ the indicator variable for the event that at least one of $\vec{e}_j \in \mathcal{B}_{\mathbb{R}^K}$ is the minimizer of $L_{\vec{\theta}}$. Then $\Pr\{\mathbb{I}_{L_{\vec{\theta}}} = 1\} < \frac{K}{\pi e^N}$, i.e. $\Pr\{\mathbb{I}_{L_{\vec{\theta}}} = 0\} \ge 1 - \frac{K}{\pi e^N}$.*

**Lemma 5.3.** *Define $\mathfrak{F}(\vec{y}, L(f)) = \left\{\vec{f} \in \mathbb{R}^N : \left\|\vec{f} - \vec{y}\right\|^2 = L_f\right\}$ for given $\vec{y}$ and $\vec{f}$. Then we have $\Pr_{\vec{f} \in \mathfrak{F}(\vec{y}, L(f))}\left\{\langle \vec{f} - \vec{y}, \vec{f} \rangle = 0\right\} < \frac{1}{\pi e^N}$.*

The above Lemmas prove Theorem 1. The following corollary is the closed from of the optimal weights.

**Corollary 5.1.** *The closed form of the minimizer is:*

$$[\theta_t]_{t\in[K]^+} = \left[\langle \vec{f}_s, \vec{f}_t \rangle\right]^{-1}_{s,t\in[K]^+} \times \left[\langle \vec{f}_s, \vec{y} \rangle\right]_{s\in[K]^+}.$$

In the following, we deal with $\sigma \circ \Theta(f_1, ..., f_K)$ and $\Theta_1 \circ \sigma \circ \Theta(f_1, ..., f_K)$.

**Theorem 2.** *Assume the set of pre-trained components $\{f_j(\mathbf{x}_j)\}_{j=1}^K$ satisfies both NPC and LIC, and $g$ be $\sigma \circ \Theta(f_1, ..., f_K)$. Then with probability at least $1 - \frac{K}{\pi e^N}$ there exists $\vec{\theta}$ s.t. $L_{\vec{\theta}}(\mathbf{x}; g) < \min_{j\in[K]} L(f_j(\mathbf{x}_j))$.*

*Proof.* (**Proof Sketch**) The whole proof is split to Lemma 5.4,5.5, and 5.6. The idea is to find an interval in the domain of $\sigma$ such that the output can approximate linear function as well as possible. Then in this interval, the activation $\sigma$ can approximate any given pre-trained component. However, under the assumptions LIC and NPC the gradient of the loss $L$ is not zero with high probability. Since the training is based on the gradient descent algorithm, this none-zero gradient leads the direction of updating process to obtain a lower loss. $\square$

**Lemma 5.4.** *Let $N, K$ and $j \in [K]$ be fixed. For small enough $\epsilon$, there exists $\vec{\theta} \in Z_{F,1,\epsilon}$ and $0 < \alpha \in \mathbb{R}$ s.t. $|\sigma \circ \Theta_{(0)}(f_1, ..., f_K) - \frac{f_j(\mathbf{x})}{\alpha}| < \epsilon$.*

**Lemma 5.5.** *Assume NPC holds with $\epsilon^* > 0$. If $\vec{\theta}_{\epsilon^*/3}$ satisfies $|\sigma \circ \Theta_{(0)}(f_1, ..., f_K)(\mathbf{x}) - f_j(\mathbf{x})| < \frac{\epsilon^*}{3N}$ for any $j \in [K]^+$, then $\nabla_{\vec{\theta}} L(\vec{\theta}_{\epsilon^*/3}) \ne \vec{0}$.*

**Lemma 5.6.** *If $\vec{\theta}_{\epsilon^*/3}$ makes $\nabla_{\vec{\theta}} L(\vec{\theta}_{\epsilon^*/3}) \ne \vec{0}$, then there exist $\vec{\theta}$ s.t. $L_{\vec{\theta}}(\mathbf{x}; g) < \min_{j\in[K]^+} L(f_j(\mathbf{x}_j))$.*

Now we consider the difference of losses of $\sigma \circ \Theta'(f_1, ..., f_K)$ and $\sigma \circ \Theta(f_1, ..., f_{K-1})$.

**Theorem 3.** *Assume the set of components $\{f_j(\mathbf{x}_j)\}_{j=1}^K$ satisfies LIC. Let $g_{K-1} = \Theta(f_1, ... f_{K-1})$ and $g_K = \Theta(f_1, ... f_K)$. With probability at least $1 - \frac{K}{\pi e^N}$ , there is a vector $\vec{\theta} \in \mathbb{R}^K \setminus \mathbb{B}_{\mathbb{R}^K}$ s.t. $L_{\vec{\theta}}(\mathbf{x}; g_K) < L_{\vec{\theta}}(\mathbf{x}; g_{K-1})$.*

Table 1: Validation Error of Image Classification

|                     | ResNet50 | SIFT    | composite |
|---------------------|----------|---------|-----------|
| number of parameter | 25636712 | 2884200 | 3000      |
| Validation Error (%) | 70.1184 | 13.1252 | 71.5079   |

*Proof.* (**Proof Sketch**) The idea is directly solve the inequality for the case of $K = 2$, and then generalize the result to larger $K$. □

The following provides a generalized error bound for a composite neural network.

**Theorem 4.** *Assume pre-trained components $\{f_j\}_{j=1}^K$ satisfy LIC and NPC. Let $\{GE(f_j)\}_{j=1}^K$ be corresponding generalization errors of $\{f_j\}_{j=1}^K$, and $\Theta_{(L)} \circ \sigma_{(L)} \circ \cdots \circ \sigma_{(1)} \circ \Theta_{(0)}(f_1, ..., f_K)$ be the composite neural network. Denote the generalization error, $\mathbf{E}\{L(\Theta_{(L)} \circ \sigma_{(L)} \circ \cdots \circ \sigma_{(1)} \circ \Theta_{(0)}(f_1, ..., f_K))\}$, of the composite neural network as $\mathbf{E}\{L_{\Theta, f_1, ..., f_K}\}$. Then with high probability, there exist a setting of $\{\Theta_{(L)}^*, ..., \Theta_{(0)}^*\}$ such that $\mathbf{E}\{L_{\Theta, f_1, ..., f_K}\} \leq \Theta_{(L)}^*(GE(f_1), ...G_E(f_K))$.*

*Proof.* (**Proof Sketch**) We apply the idea similar to Kawaguchi (2016): the exception of non-liner activations is same with the exception of liner activations. Previous theorems provide that with high probability there exists the solution of $\Theta_{(i)}, \forall i \in [L]^+$ s.t. each $\Theta_{(i)+1}\sigma\Theta_{(i)}$ approximates a degree one polynomial $A_{\Theta_{(i)+1}\sigma\Theta_{(i)},1}$ as well as possible. If the weights are obey the normal distribution, then $\mathbf{E}\{L_{\Theta, f_1, ..., f_K}\} \leq \Theta_{(L)}^*(GE(f_1), ...G_E(f_K))$. □

## 6 EMPIRICAL STUDIES

This section is to numerically verify the performance of composite network for two distinctively different applications, image classification and PM2.5 prediction. For image classification, we examined two pre-trained components, the ResNet50 (He et al., 2016) from Keras and the SIFT algorithm(Lowe, 1999) from OpenCV, running on the benchmark of ImageNet competition(Russakovsky et al., 2015). For PM2.5 prediction, we implemented several models running on the open data of local weather bureau and environment protection agency to predict the PM2.5 level in the future hours.

### 6.1 IMAGENET CLASSIFICATION

We chose Resnet50 as the pre-trained baseline model and the SIFT model as an auxiliary model to form a composite neural network to validate the proposed theory. The experiments are conducted on the 1000-class single-label classification task of the ImageNet dataset, which has been a well received benchmark for image classification applications. A reason to choose the SIFT (Scale-Invariant Feature Transform) algorithm is that its function is very different from ResNet and it is interesting to see if the performance of ResNet50 can be improved as predicted from our theory.

We trained the SIFT model using the images of ImageNet, and directed the output to a CNN to extract useful features before merging with ResNet50 output. In the composite model, the softmax functions of both ResNet50 and SIFT model are removed that the outputs of length 1000 of both models are merged before the final softmax stage. During the training process of composite network, the weights of ResNet50 and SIFT model are fixed, and only the connecting weights and bias are trained.

The ResNet50 was from He et al. that its Top-1 accuracy in our context was lower than reported in (He et al., 2016) since we did not do any fine tuning and data preprocessing. In the Figure 1, it shows the composite network has higher accuracy than ResNet50 during almost the complete testing run. Table 1 shows the same result that the composite network performs better too. The experiment results support the claims of this work that a composite network performs better than any of its components, and more components work better than less components.

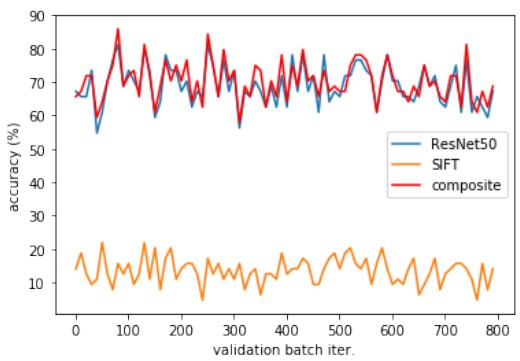

Figure 1: Image Classification Validation Accuracy

## 6.2 PM2.5 PREDICTION

The PM2.5 prediction problem is to forecast the particle density of fine atmospheric matter with the diameter at most 2.5 $\mu m$ (PM2.5) in the future hours, mainly, for the next 12, 24, 48, 72 hours. The datasets used are open data provided by two sources including Environmental Protection Administration (EPA)[1] , and Center Weather Bureau (CWB)[2]. The EPA dataset contains 21 observed features, including the speed and direction of wind, temperature, relative humidity, PM2.5 and PM10 density, etc., from 18 monitoring stations, with one record per hour. The CWB has seventy monitoring stations, one record per 6 hours, containing 26 features, such as temperature, dew point, precipitation, wind speed and direction, etc. We partitioned the observed area into a grid of 1140 km$^2$ with 1 km$\times$1 km blocks and aligned the both dataset into one-hour period. We called the two datasets as air quality and weather condition dataset.

We selected ConvLSTM (Convolution LSTM) and FNN (fully connected neural network) as the components used in this experiment. The reason to select ConvLSTM is that the dispersion of PM2.5 is both spatially and temporally dependent and ConvLSTM is considered capable of catching the dependency, and FNN is a fundamental neural network that acts as the auxiliary component in the experiment.

The prediction models were trained with the data of 2014 and 2015 years, then the 2016 data was used for testing. We considered two function compositions, the linear combination $\Theta$ and the Logistic function $\sigma_1$ (as Theorem 2), to combine the two components to examine the applicability of the proposed theorems.

We trained and tested both ConvLSTM and FNN using air quality dataset (Dataset A) and weather condition dataset (Dataset B) separately as the baselines (denoted as $f_1$, $f_2$, $f_3$ and $f_4$) and their training error and testing error in MSE are list in the first part of Table 2. Then we composited FNNs using Dataset A and Dataset B, each FNN can be pre-trained (denoted as x) or non-instantiated (denoted as o). In addition, we used both linear and Sigmoid activation functions. As a result, we had eight combinations, as list in the part two. We treated ConvLSTM in the same way and the outcomes were in the part 3. Finally, we composited using one FNN and one ConvLSTM that each was the best in their category, and the resulting composite network was a tree of depth 2. For instance, the candidate of ConvLSTM of part 4 for 12 hours prediction was the 4th row (i.e., $\Theta(f_3^o, f_4^o)$) of part 3. Their training and testing errors in MSE were listed in the part 4.

From the empirical study results, it shows mostly the proposed theorems are followed. While the composite networks with all pre-trained components may not perform better than others in their category, (which is not a surprise), what we expect to see is after adding a new component, the composite network has improvement over the previous one. For example, the $\sigma \circ \Theta(f_3^\times, f_4^\times)$ has strictly better accuracy than both $f_3$ and $f_4$ for all future predictions. Another example is the NEXT 48 hr, $\sigma \circ \Theta(C^\times, F^\times)$ also has strictly better accuracy than both $C = \sigma \circ \Theta(f_3^o, f_4^o)$ and $F = \sigma \circ \Theta(f_3^o, f_4^o)$.

---

[1]https://opendata.epa.gov.tw/Home

[2]http://opendata.cwb.gov.tw/index

Table 2: Training and Testing Errors of PM2.5 Prediction

| Model | Next 12 hr | | Next 24 hr | | Next 48 hr | | Next 72 hr | |
|---|---|---|---|---|---|---|---|---|
| (input) | TarinError | TestError | TarinError | TestError | TarinError | TestError | TarinError | TestError |
| $\mathbf{f_1}$: FNN-A | 100.1812 | 92.8528 | 134.7095 | 118.6065 | 141.6287 | 136.8358 | 148.1807 | 143.9980 |
| $\mathbf{f_2}$: FNN-B | 134.1137 | 120.0019 | 139.8016 | 128.5960 | 136.7693 | 134.9001 | 142.7637 | 140.8650 |
| $\mathbf{f_3}$: ConvLSTM-A | 54.2775 | 88.8156 | 57.5677 | 111.9122 | 74.8937 | 129.7418 | 77.7394 | 132.8923 |
| $\mathbf{f_4}$: ConvLSTM-B | 67.8625 | 118.4351 | 73.1519 | 125.6062 | 68.7069 | 137.0789 | 84.9656 | 138.6642 |
| $\Theta(f_1^\times,f_2^\times)$ | 99.7005 | $^{F:}$90.0214 | 130.7800 | 115.9283 | 139.9744 | $^{F:}$132.4764 | 144.6826 | $^{F:}$137.8403 |
| $\Theta(f_1^\times,f_2^\circ)$ | 95.6804 | 93.0173 | 120.3185 | 117.9781 | 134.3893 | 134.0270 | 139.6226 | 140.5209 |
| $\Theta(f_1^\circ,f_2^\times)$ | 95.8110 | 93.1131 | 121.9737 | 117.7771 | 134.0676 | 135.2255 | 136.2009 | 144.0116 |
| $\Theta(f_1^\circ,f_2^\circ)$ | 101.1584 | 90.2671 | 126.6807 | 114.5264 | 132.6726 | 132.8069 | 139.2339 | 139.3322 |
| $\sigma\circ\Theta(f_1^\times,f_2^\times)$ | 102.7556 | 90.6280 | 133.1453 | 117.7397 | 135.9256 | 133.2544 | 145.1052 | 139.2513 |
| $\sigma\circ\Theta(f_1^\times,f_2^\circ)$ | 98.1241 | 93.1098 | 127.4999 | 118.8107 | 135.1553 | 134.1469 | 137.7562 | 142.1778 |
| $\sigma\circ\Theta(f_1^\circ,f_2^\times)$ | 94.9931 | 91.4667 | 124.5461 | 117.7332 | 131.2684 | 135.1281 | 140.1604 | 144.5220 |
| $\sigma\circ\Theta(f_1^\circ,f_2^\circ)$ | 98.2596 | 91.3646 | 124.4182 | $^{F:}$114.2274 | 134.5078 | 132.8316 | 138.0456 | 139.8351 |
| $\Theta(f_3^\times,f_4^\times)$ | 27.1760 | 85.8922 | 49.1624 | 108.3157 | 37.2116 | 123.2186 | 60.6415 | 131.0565 |
| $\Theta(f_3^\times,f_4^\circ)$ | 27.1519 | 81.7688 | 42.7932 | 104.2375 | 33.2831 | $^{C:}$110.4213 | 74.3055 | $^{C:}$110.9952 |
| $\Theta(f_3^\circ,f_4^\times)$ | 27.4436 | 78.5360 | 44.3214 | 107.0898 | 31.4910 | 129.1829 | 68.4413 | 139.5661 |
| $\Theta(f_3^\circ,f_4^\circ)$ | 26.3063 | $^{C:}$70.8670 | 40.1879 | 100.8474 | 25.3312 | 119.1634 | 76.1782 | 120.6814 |
| $\sigma\circ\Theta(f_3^\times,f_4^\times)$ | 28.3844 | 84.9029 | 43.2981 | 109.3709 | 31.2413 | 123.0041 | 63.7286 | 130.4122 |
| $\sigma\circ\Theta(f_3^\times,f_4^\circ)$ | 27.5981 | 80.7848 | 44.4197 | 98.1051 | 29.7649 | 111.6793 | 69.3182 | 117.0719 |
| $\sigma\circ\Theta(f_3^\circ,f_4^\times)$ | 26.4125 | 78.3990 | 42.3181 | 103.3361 | 30.4183 | 128.4138 | 64.4193 | 136.6043 |
| $\sigma\circ\Theta(f_3^\circ,f_4^\circ)$ | 26.5131 | 75.5778 | 42.3912 | $^{C:}$94.6242 | 27.5812 | 112.8075 | 70.5132 | 117.6480 |
| $\Theta(C^\times,F^\times)$ | 26.6556 | 70.1159 | 34.6885 | 92.2737 | 29.6484 | 107.6833 | 52.0060 | $^{H:}$110.1283 |
| $\Theta(C^\times,F^\circ)$ | 24.2349 | 67.4414 | 31.7202 | 90.7795 | 30.1328 | $^{H:}$105.1804 | 46.9994 | 111.2227 |
| $\Theta(C^\circ,F^\times)$ | 22.7651 | 75.9468 | 24.2132 | 96.3126 | 24.4488 | 114.1803 | 49.0663 | 117.2747 |
| $\Theta(C^\circ,F^\circ)$ | 21.9103 | 68.0660 | 20.9072 | 91.9323 | 23.2868 | 112.8605 | 30.6875 | 113.6968 |
| $\sigma\circ\Theta(C^\times,F^\times)$ | 26.5950 | 69.1897 | 34.7400 | 92.4715 | 29.9215 | 108.7482 | 52.3708 | 111.5474 |
| $\sigma\circ\Theta(C^\times,F^\circ)$ | 24.0223 | $^{H:}$66.4733 | 28.8401 | $^{H:}$90.7257 | 28.5033 | 108.8896 | 46.2711 | 110.1613 |
| $\sigma\circ\Theta(C^\circ,F^\times)$ | 22.4443 | 83.5953 | 22.5040 | 96.4027 | 28.4714 | 112.0727 | 35.3947 | 114.5947 |
| $\sigma\circ\Theta(C^\circ,F^\circ)$ | 38.4899 | 67.1819 | 17.6041 | 92.2343 | 33.9710 | 105.7977 | 40.2934 | 110.3585 |
| Let $\mathbf{f_5}$ be non-instantiated CNN with future rain fall input. | | | | | | | | |
| $\sigma\circ\Theta(H^\times,f_5^\circ)$ | 24.1487 | 65.5776 | 30.6243 | 87.2777 | 55.9261 | 102.2878 | 50.5158 | 108.8087 |
| $\sigma\circ\Theta(H^\circ,f_5^\circ)$ | 58.0852 | 68.8111 | 22.8776 | 90.2324 | 39.0996 | 112.1639 | 36.0659 | 109.8240 |

Part 1: pre-trained components $f_i$, $i \in [4]$. Part 2: composite $f_1$ and $f_2$ by linear $\Theta(\cdot)$ or logistic $\sigma\circ\Theta(\cdot)$; similar for Part 3-5
$^\times$: un-trainable component, i.e. pre-trained. $^\circ$: trainable component (original weights was deleted).
The best model of Part 2 (/Part 3) was assigned as composite model $F$ (/$C$) which will be used in Part 4.
The best model of Part 4 was assigned as composite model $H$ which will be used in Part 5.

# 7 CONCLUSION

In this work, we investigated the composite neural network with pre-trained components problem and showed that the overall performance of a composite neural network is better than any of its components, and more components perform better than less components. In addition, the developed theory consider all differentiable activation functions.

While the proposed theory ensures the overall performance improvement, it is still not clear how to decompose a complicated problem into components and how to construct them into a composite neural network in order to have an acceptable performance. Another problem worth some thinking is when the performance improvement will diminish (by power law or exponentially decay) even adding more components. However, in the real world applications, the amount of data, data distribution and data quality will highly affect the performance.

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

## 8 Supplementary material

For self-contained, we list some common Taylor expansion in the following.

Logistic: $S(z) := \frac{1}{1+e^{-z}} = \frac{1}{2} + \frac{1}{4}z - \frac{1}{48}z^3 + \frac{1}{480}z^5 - \frac{17}{80640}z^7 + O(z^9), \forall z \in \mathbb{R}$,

Hyperbolic Tan: $tanH(z) = \frac{e^z - e^{-z}}{e^z + e^{-z}} = z - \frac{1}{3}z^3 + \frac{2}{15}z^5 + O(z^7), \forall |z| \leq \frac{\pi}{2}$

arcTan: $arctan(z) = z - \frac{1}{3}z^3 + \frac{1}{5}z^5 + +O(z^7), \forall |z| \leq 1$.

**Definition 1.** *Given an activation $\sigma(z)$ and its Taylor expansion $T_\sigma(z)$, let $A_{\sigma,D}(z)$ be the truncated the monomials of degree at most $D$ from $T_\sigma(z)$. We define $A_{\sigma,D}(z)$ as the $D$-degree Taylor approximation polynomial, and $R_{\sigma,D+1}(z)$ as the remainder part such that $T_\sigma(z) = A_{\sigma,D}(z) + R_{\sigma,D+1}(z)$.*

For instance, if we set $D = 3$ then the Taylor expansion of Logistic function $S(z)$ is separated as the approximation part $A_{S(z),3}(z) = \frac{1}{2} + \frac{1}{4}z - \frac{1}{48}z^3$ and the remainder part $R_{S(z),4}(z) = \frac{1}{480}z^5 + O(z^7)$.

**Proposition 8.1.** *(Error Bound of The Remainder)*
*Let $S(z)$ be the Logistic function. Consider the approximation $A_{S(z),\leq D}(z)$ and the remainder $R_{S(z),D+1}(z)$ defined as above. For given $\epsilon \in (0, \frac{1}{1000})$ and $D \in \mathbb{N}$, if $|z| < \epsilon^{1/(D+2)}$, then $|S(z) - A_{S(z),D}(z)| = |R_{S(z),D+1}(z)| < \epsilon$.*

*Proof.* Note that if $\epsilon < 1$ then for all $D \in \mathbb{N}$, $\epsilon^{1/(D+1)} < 1$. If $|z| < \epsilon^{1/3}$ and $D = 1$, then

$$|R_{S(z),D+1}(z)| \leq \left| -\frac{1}{48}z^3 + \frac{1}{480}z^5 - \frac{17}{80640}z^7 + O(z^9) \right| < \frac{1}{24}\epsilon < \epsilon.$$

The general case ($D \geq 2$) can be proven by the same argument as above. □

This Proposition means that for a suitable range of $z$, the Logistic function can be seen as a linear function with the error at most $\epsilon$.

**Definition 2.** *For the Logistic activation $\sigma(z) = S(z)$, $\epsilon > 0$ and given polynomial degree $D$, we define $\mathcal{Z}_{D,\epsilon} = \{z \in \mathbb{R} : |\sigma(z) - A_{\sigma,D}(z)| < \epsilon\}$. Furthermore, for given components $\{f_j : j \in [K]\} = F$, we consider the variable $z = \Theta(f_1, ..., f_K)$ and define*

$$\mathcal{Z}_{F,D,\epsilon} = \left\{ \vec{\theta} \in \mathbb{R}^{K+1} : z = \Theta(f_1, ..., f_K), |\sigma(z) - A_{\sigma,D}(z)| < \epsilon \right\}.$$

Observe that if the parameters $\epsilon$, $F$, and $|F| = K$ are fixed, then $\mathcal{Z}_{F,D,\epsilon} \subset \mathcal{Z}_{F,D+1,\epsilon} \subset \mathbb{R}^{K+1}$.

### 8.1 Function Composition by Linear Combination

Recall that for a set of pre-trained components $\{f_j(\mathbf{x}_j) : j \in [K]\}$, $\Theta_{(0)}(f_1, ...f_K) = \sum_{j \in [K]^+} \theta_{0,j} f_j$, where $f_0 = 1$. For simplicity, we consider $\Theta_{(1)}(\mathbf{z}) = \alpha \mathbf{z}$. This means $\Theta_{(1)} \circ \sigma \circ \Theta_{(0)}(f_1, ...f_K) = \theta_{1,1}\sigma\left(\sum_{j \in [K]^+} \theta_{0,j} f_j\right) + \theta_{1,0}$.

Theorem 1 is a consequence of the following lemmas:

*Proof.* (of Lemma 5.1)
For simplicity of notations, let $g(\mathbf{x}) = \Theta_{(0)}(f_1, ...f_K)$, hence $g(\mathbf{x}) = \sum_{j \in [K]^+} \theta_j f_j(\mathbf{x}_j)$. Also recall that $L_{\vec{\theta}}(\mathbf{x}; g) = \sum_{i=1}^{N} \left(g(\mathbf{x}^{(i)}) - \mathbf{y}^{(i)}\right)^2$. To prove the existence of the minimizer, it is enough to solve the equations of critical points, in the case of a quadratic object function. That is, to solve the set of equations:

$$\nabla_{\vec{\theta}} L(\mathbf{x}; g) = \left(\frac{\partial L}{\partial \theta_0}, \ldots, \frac{\partial L}{\partial \theta_K}\right)^T = (0, \ldots, 0)^T,$$

where for each $s, t \in [K]^+$,

$$
\begin{aligned}
\frac{\partial L}{\partial \theta_s} &= 2 \sum_{i=1}^{N} \left( g(\mathbf{x}^{(i)}) - \mathbf{y}^{(i)} \right) \cdot f_s(\mathbf{x}^{(i)}) = 2 \sum_{i=1}^{N} \left( \sum_{j \in [K]^+} \theta_j f_j(\mathbf{x}_j^{(i)}) - \mathbf{y}^{(i)} \right) \cdot f_s(\mathbf{x}^{(i)}) \\
&= 2 \left( \sum_{j \in [K]^+} \theta_j \langle \vec{f_s}, \vec{f_j} \rangle - \langle \vec{f_s}, \vec{y} \rangle \right).
\end{aligned}
$$

Hence, to solve $\nabla_{\vec{\theta}} L(\mathbf{x}; g) = \vec{0}$ is equivalent to solve $\left[ \langle \vec{f_s}, \vec{f_t} \rangle \right]_{s,t \in [K]^+} \times [\theta_t]_{t \in [K]^+} = \left[ \langle \vec{f_s}, \vec{y} \rangle \right]_{s \in [K]^+}$, where $\left[ \langle \vec{f_s}, \vec{f_t} \rangle \right]_{s,t \in [K]^+}$ is a $(K+1)$ by $(K+1)$ matrix, $[\theta_t]_{t \in [K]^+}$ and $\left[ \langle \vec{f_s}, \vec{y} \rangle \right]_{s \in [K]^+}$ are both 1 by $(K+1)$.

Note that linear independence of $\{\vec{f_j}\}_{j \in [K]^+}$ makes $\left[ \langle \vec{f_s}, \vec{f_t} \rangle \right]_{s,t \in [K]^+}$ a positive-definite Gram matrix (Horn & Johnson, 2012), which means the inversion $\left[ \langle \vec{f_s}, \vec{f_t} \rangle \right]_{s,t \in [K]^+}^{-1}$ exists. Then $\vec{\theta}$ is solved:

$$
[\theta_t]_{t \in [K]^+} = \left[ \langle \vec{f_s}, \vec{f_t} \rangle \right]_{s,t \in [K]^+}^{-1} \times \left[ \langle \vec{f_s}, \vec{y} \rangle \right]_{s \in [K]^+} \tag{2}
$$

The above shows the existence of the critical points. On the other hand, since $L_{\vec{\theta}}(\mathbf{x}; g)$ is the summation of square terms, i.e. paraboloid, the the critical points can only be the minimum. □

The meaning of the gradient on a function surface is the direction that increases the function value most efficiently. Hence, if the gradient is not the zero vector then the corresponding point can not be the minimizer of the function surface. Recall for any $s \in [k]$,

$$
\left[ \frac{\partial L}{\partial \theta_t} \right]_{t \in [K]^+} \big|_{\vec{\theta} = \vec{e}_s} = 2 \left[ \langle \vec{f_s} - \vec{y}, \vec{f_t} \rangle \right]_{t \in [K]^+}.
$$

Before the proof of Lemma 5.2, we need the upper bound of the probability of some events. Note that $\vec{y}$ is defined according to the given training data, and for each $j \in [K]^+$ the length of $\vec{f_j} - \vec{y}$, i.e. $\left\| \vec{f} - \vec{y} \right\|$, is also given. The question is, for fixed $\vec{y}$ what is the probability of selected $\vec{f}$ is perpendicular to $\vec{f} - \vec{y}$? A folklore approach is considering that $\{\vec{f} = (f(\mathbf{x}^{(1)}), ..., f(\mathbf{x}^{(N)}))\}$ obeys the normal distribution, and setting the mean of $f(\mathbf{x}^{(i)})$ as $\mathbf{y}^{(i)}$ for each $i \in [N]$. In the following we propose another simple probability argument to obtain a loose upper bound.

*Proof.* (of Lemma 5.3) Observe that $\langle \vec{f} - \vec{y}, \vec{f} \rangle = 0 \Leftrightarrow (\vec{f} - \vec{y}) \perp \vec{f}$, which implies the angle between them, $\angle_{(\vec{f} - \vec{y}), \vec{f}}$, is in the interval $[\frac{\pi - \epsilon}{2}, \frac{\pi + \epsilon}{2}]$ for small $\epsilon \in \mathbb{R}^+$, as shown in the left part of Figure 2. The red, orange, and blue vectors show three possibles of the pair of $\vec{f}$ and $\vec{f} - \vec{y}$. The length of $\vec{f} - \vec{y}$ is fixed since $\vec{f}$ and $\vec{y}$ are given, but the angle between $\vec{f} - \vec{y}$ and $\vec{y}$ can decide whether $(\vec{f} - \vec{y}) \perp \vec{f}$. The gray circle collects all possible end-point of the vector $\vec{f} - \vec{y}$ emission from the end-point of $\vec{y}$. Although on the whole circle there are exactly two specific angles [3] can satisfy $(\vec{f} - \vec{y}) \perp \vec{f}$, we give a loose small interval $\epsilon$ with respect to $\pi$. In particularly, we set $0 < \epsilon < e^{-N}$.

$$
\Pr_{\vec{f} \in \mathfrak{F}(\vec{y}, L_f)} \left\{ \angle_{(\vec{f} - \vec{y}), \vec{f}} = \frac{\pi}{2} \right\} \leq \Pr_{\vec{f} \in \mathfrak{F}(\vec{y}, L_f)} \left\{ \frac{\pi - \epsilon}{2} \leq \angle_{(\vec{f} - \vec{y}), \vec{f}} \leq \frac{\pi + \epsilon}{2} \right\} = \frac{\epsilon}{\pi} < \frac{1}{\pi e^N}.
$$

□

Now we are ready to proof Lemma 5.2.

---

[3] That is, two points on the circumference, which is in fact measure zero on all possible angle $[0, 2\pi)$

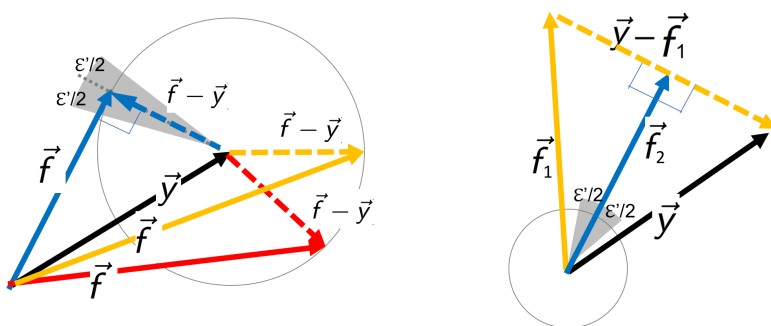

Figure 2: An illustration of Lemma 5.3 (the left) and Lemma 5.4 (the right)

*Proof.* (of Lemma 5.2)
We denote $\mathcal{A}$ the event that at least one of $\vec{e}_j \in \mathcal{B}_{\mathbb{R}^K}$ is the minimizer of $L(\vec{\theta})$ for convenience.

$$\mathbb{I}_{L(\vec{w})} = 1 \Leftrightarrow \text{the event } \mathcal{A} \text{ is true}$$
$$\Rightarrow \left[\frac{\partial L}{\partial \theta_t}\right]_{t \in [K]^+} \big|_{\vec{\theta} = \vec{e}_s} = 2\left[\langle \vec{f}_s - \vec{y}, \vec{f}_t \rangle\right]_{t \in [K]^+} = [0]_{K \times 1} \text{ for some } s \in [K]^+$$
$$\Rightarrow \langle \vec{f}_1 - \vec{y}, \vec{f}_1 \rangle = 0 \wedge \langle \vec{f}_1 - \vec{y}, \vec{f}_2 \rangle = 0 \wedge \cdots \wedge \langle \vec{f}_1 - \vec{y}, \vec{f}_K \rangle = 0$$
$$\text{or } \cdots \text{ or } \langle \vec{f}_K - \vec{y}, \vec{f}_1 \rangle = 0 \wedge \langle \vec{f}_K - \vec{y}, \vec{f}_2 \rangle = 0 \wedge \cdots \wedge \langle \vec{f}_K - \vec{y}, \vec{f}_K \rangle = 0$$

Hence, for given $\vec{y}$ and $L(f_j) = \left\|\vec{f}_j - \vec{y}\right\|^2, \forall \in [K]^+$, we have

$$\Pr\left\{\mathbb{I}_{L(\vec{w})} = 1\right\} \leq \sum_{j \in [K]^+} \Pr\left\{\langle \vec{f}_j - \vec{y}, \vec{f}_1 \rangle = 0 \wedge \langle \vec{f}_j - \vec{y}, \vec{f}_2 \rangle = 0 \wedge \cdots \wedge \langle \vec{f}_j - \vec{y}, \vec{f}_K \rangle = 0\right\}$$
$$\leq K \cdot \Pr\left\{\langle \vec{f}_1 - \vec{y}, \vec{f}_1 \rangle = 0\right\}$$
$$< \frac{K}{\pi e^N},$$

where the second inequality is based on the symmetry between $\vec{f}_s$ and $\vec{f}_t$ for any $s, t \in [K]^+$, and the last inequality is by Lemma 5.3. □

*Proof.* (of Theorem 3) We start from a simple case:
Claim: $\exists \beta \in \mathbb{R}$ s.t.

$$\sum_{i \in [N]} (f_1(x_i) - y_i)^2 - \sum_{i \in [N]} (f_1(x_i) + \beta f_2(x_i) - y_i)^2 > 0.$$

*Proof.*

$$\sum_{i \in [N]} (f_1(x_i) - y_i)^2 - \sum_{i \in [N]} (f_1(x_i) + \beta f_2(x_i) - y_i)^2$$
$$= \sum_{i \in [N]} \left[(f_1(x_i) - y_i)^2 - (f_1(x_i) + \beta f_2(x_i) - y_i)^2\right]$$
$$= \left[-\sum_{i \in [N]} f_1(x_i)^2\right] \beta^2 + \left[2 \sum_{i \in [N]} (f_2(x_i)y_i - f_2(x_i)f_1(x_i))\right] \beta$$

Observe that the above is a quadratic equation of $\beta$ with negative leading coefficient. Hence, to obtain the maximum of the difference, we can set

$$\beta = \frac{\sum_{i \in [N]} (f_2(x_i)y_i - f_2(x_i)f_1(x_i))}{\sum_{i \in [N]} f_1(x_i)^2} = \frac{\langle \vec{y} - \vec{f}_1, \vec{f}_2 \rangle}{\langle \vec{f}_2, \vec{f}_2 \rangle}$$

Note that if $\langle \vec{y} - \vec{f}_1, \vec{f}_2 \rangle = 0$ then the last pre-trained component is no need to be added. We aim to calculate the probability of this case. Observe that $\langle \vec{y} - \vec{f}_1, \vec{f}_2 \rangle = 0 \Leftrightarrow (\vec{y} - \vec{f}_1) \perp \vec{f}_2$. This condition is different from previous Lemma. Here we have to find the upper bound of the probability of $(\vec{y} - \vec{f}_1) \perp \vec{f}_2$ for given $\vec{f}_1$ and $\vec{y}$. As shown in the left part of Figure 2), the angle between $\vec{f}_2$ and $\vec{y}$ must be in a specific interval, say $[\frac{\pi - \epsilon}{2}, \frac{\pi + \epsilon}{2}]$ for small $\epsilon \in \mathbb{R}^+$. In order to be concrete, we set $0 < \epsilon < e^{-N}$.

$$\Pr_{\vec{f} \in \mathfrak{F}(\vec{y}, 1)} \left\{ (\vec{y} - \vec{f}_1) \perp \vec{f}_2 \right\} \leq \frac{\epsilon}{\pi} < \frac{1}{\pi e^N}.$$

$\square$

The general case can be reduced to the above claim by considering $g_{K-1}$ as $f_1$ and $\theta_k f_K$ as $\beta f_2$. Furthermore, since there there $K$ possibles the be selected as the least pre-trained component, the probability is upper bounded by $\frac{K}{\pi e^N}$.

$\square$

## 8.2 Function Composition by Non-Linear Activation

*Proof.* (of Lemma 5.4) Although the lemma is an existence statement, we give a constructive proof here. By setting $D = 1$ in Proposition 8.1, we know that for Logistic $S(z)$ and $0 < \epsilon < 1/1000$, the degree-one Taylor approximation $A_{S(z),1} = \frac{1}{2} + \frac{1}{4}z$ with the remainder $|R_{S(z),2}| < \epsilon$. Define $M := 10 \cdot \max_{j \in [K]^+, i \in [N]} \{|f_j(x_i)|\}$. Hence by setting $z = \frac{f_j(\mathbf{x}_j)}{M}$, we have $\left| S\left( \frac{f_j(\mathbf{x}_j)}{M} \right) - \frac{1}{2} - \frac{f_j(\mathbf{x}_j)}{4M} \right| < \epsilon$. This means that for the given $j \in [K]$, $\theta_j = \frac{1}{M}$, $\theta_0 = $ and for all $j' \neq j$, $\theta_{j'} = 0$. Furthermore, $\alpha = 4M$.

$\square$

This lemma implies that the $S(z)$ can approximate linear function as possible in a non-zero length interval, hence if the scaling of $\vec{\theta}$ is allowed then Theorem 1 can be applied.

**Corollary 8.1.** *If the activation function is the Logistic, $S(z)$, and $\{f_j(\mathbf{x}_j)\}_{j=1}^K$ satisfies LIC, then with high probability there is a vector $\vec{\theta}$ s.t. $L_{\vec{\theta}}\left( \mathbf{x}; \sigma \circ \Theta_{(0)}(f_1, ..., f_K) \right) < \min_{j \in [K]^+} L(f_j(\mathbf{x}_j))$.*

*Proof.* Set $\epsilon$ of above Lemma as $\frac{\epsilon^*}{3N}$, then previous Lemma shows that there exists $\theta$ which maps $\{f_j\}$ into $\mathcal{Z}_{\sigma \circ \Theta_{(0)}, 1, \frac{\epsilon^*}{3N}}$. Since the output of $\sigma \circ \Theta_{(0)}$ is a linear function with error at most $\frac{\epsilon^*}{3N}$, we have the same conclusion.

$\square$

*Proof.* Let $g(\mathbf{x}) = \sigma \circ \Theta_{(0)}(f_1, ..., f_K)(\mathbf{x})$ for short. First observe that $|g(\mathbf{x}) - f_j(\mathbf{x})| < \frac{\epsilon^*}{3N} \Rightarrow \forall i \in [N], g(\mathbf{x}^{(i)}) - f_j(\mathbf{x}^{(i)}) > \frac{-\epsilon^*}{3N}$. Then

$$\sum_{i \in [N]} \left( g(\mathbf{x}^{(i)}) - \mathbf{y}^{(i)} \right) = \sum_{i \in [N]} \left\{ (g(\mathbf{x}^{(i)}) - f_j(\mathbf{x}^{(i)})) + (f_j(\mathbf{x}^{(i)}) - \mathbf{y}^{(i)}) \right\} > N \cdot \frac{-\epsilon^*}{3N} + \epsilon^* = \frac{2\epsilon^*}{3} > 0.$$

On the other hand, it can be calculated that

$$\nabla_{\vec{\theta}} L\left( \mathbf{x}; g \right) |_{\vec{\theta} = \vec{\theta}_{\epsilon^*/3}} = \left( \frac{\partial L}{\partial w_0}, \frac{\partial L}{\partial \theta_1}, ..., \frac{\partial L}{\partial \theta_K}, \frac{\partial L}{\partial b_0} \right)^T |_{\vec{\theta} = \vec{\theta}_{\epsilon^*/3}},$$

where $[a]^T$ is the transpose of the matrix $[a]$. Also note that $\frac{\partial L}{\partial w_0}|_{\vec{\theta} = \vec{\theta}_{\epsilon^*/3}} = 2 \cdot \sum_{i \in [N]} (g(\mathbf{x}^{(i)}) - \mathbf{y}^{(i)}) \cdot S(z)$, and $\frac{\partial L}{\partial b_0}|_{\vec{\theta} = \vec{\theta}_{\epsilon^*/3}} = 2 \cdot \sum_{i \in [N]} (g(\mathbf{x}^{(i)}) - \mathbf{y}^{(i)}) \cdot 1$. Since $\sum_{i \in [N]} (g(\mathbf{x}^{(i)}) - \mathbf{y}^{(i)}) > 0$, we can conclude $\nabla_{\vec{\theta}} L\left( \mathbf{x}; g \right) |_{\vec{\theta} = \vec{\theta}_{\epsilon^*/3}} \neq \vec{0}$.

$\square$

*Proof.* (of Lemma 5.6 ) By previous Lemma, it is valid to consider the best performance component, $f_{j^*}$; i.e. $L(f_{j^*}) = \min_{j \in [K]^+} L(f_j(\mathbf{x}_j))$. Since $\nabla_{\vec{\theta}} L(\vec{\theta}_{\epsilon^*/3}) \neq \vec{0}$, by definition of the gradient, moving along the direction of $-\nabla_{\vec{\theta}} L(\vec{\theta}_{\epsilon^*/3}) / \left\| \nabla_{\vec{\theta}} L(\vec{\theta}_{\epsilon^*/3}) \right\|$ with a step-size $\alpha > 0$ must strictly decrease the value of $L_{\vec{\theta}}(\mathbf{x}; g)$. W.L.O.G, we can assume this $\alpha$ is optimal; that is, if $\alpha > r > 0$ then $L(\vec{\theta}_{\epsilon^*/3}) > L(\vec{\theta}_{\epsilon^*/3}) - r \cdot \nabla_{\vec{\theta}} L(\vec{\theta}_{\epsilon^*/3}) / \left\| \nabla_{\vec{\theta}} L(\vec{\theta}_{\epsilon^*/3}) \right\|$, while if $r = \alpha + \delta$ for some $\delta > 0$ then

$L(\vec{\theta}_{\epsilon^*/3}) \leq L(\vec{\theta}_{\epsilon^*/3}) - r \cdot \nabla_{\vec{\theta}} L(\vec{\theta}_{\epsilon^*/3}) / \left\| \nabla_{\vec{\theta}} L(\vec{\theta}_{\epsilon^*/3}) \right\|$. The issue is how to find a proper step-size $r > 0$? We consider the line search approach such that:

$$r^* = \arg\min_{r \in \mathbb{R}} \left\{ L \left( \vec{\theta}_0 - r \cdot \frac{\nabla_{\vec{\theta}} L(\vec{\theta}_0)}{\left\| \nabla_{\vec{\theta}} L(\vec{\theta}_0) \right\|} \right) \right\}$$

This outputs $F^*$ then we can make sure that $L\left(\vec{\theta}_0\right) > L\left(\vec{\theta}_0\right) - r \cdot \nabla_{\vec{\theta}} L(\vec{\theta}_0) / \left\| \nabla_{\vec{\theta}} L(\vec{\theta}_0) \right\|$. Since the underlining $\theta_0$ is $\vec{\theta}_{\epsilon^*/3}$ that makes the loss is the same with the best one. Hence, we have $(\vec{\theta}_0 - r^* \cdot \nabla_{\vec{\theta}} L(\vec{\theta}_{\epsilon^*/3}) / \left\| \nabla_{\vec{\theta}} L(\vec{\theta}_{\epsilon^*/3}) \right\|)$ to fit our goal (beating the best one). $\qquad \square$

Combining these lemmas can obtain the conclusion of Theorem 2.

**Corollary 8.2.** *The process of Lemma 5.5 converges.*

*Proof.* (of 8.2) It is know that if a monotone decreasing sequence is bounded below, then this sequence convergences. ..... (revising) Repeating the process of Lemma 2, we can obtain a strictly decreasing sequence: $L(\vec{\theta}_0) > L(\vec{\theta}_1) > L(\vec{\theta}_2) > \dots$. Note that $\forall i, (\vec{\theta}_i) \geq 0$. This means the sequence is monotone decreasing and bounded below, so theoretically it converges by monotone convergence theorem of mathematical analysis. Algorithmically, the gradient descent based sequence finding process stops at some term with $\nabla_{\vec{\theta}} L(\vec{\theta}') = 0$, which is a (local) minimum or a saddle point. $\qquad \square$

**Corollary 8.3.** *If the assumptions in Theorem 2 are satisfied, then with height probability there exists $\vec{\theta}'$ s.t. $L_{\vec{\theta}'}$ is a (local) minimum or a saddle point, while $L_{\vec{\theta}'}(\mathbf{x}; g) < \min_{j \in [K]^+} |L(f_j(\mathbf{x}_j))|$ still holds.*

Assume the pre-trained component set $\{f_j(\mathbf{x}_j)\}_{j=1}^K$ satisfies both NPC and LIC, then there exists $\vec{\theta}$ s.t. $L_{\vec{\theta}}(\mathbf{x}; g) < \min_{j \in [K]^+} L(f_j(\mathbf{x}_j))$.

In the previous proof, the critical properties of activations are local linearity and differentiability. Hence, it is not hard to check that if we replace $\sigma(\cdot)$ in Eq. (1) with other common activations, the conclusion still holds. By local linearity we mean that on a non-zero length interval in its domain, the function can approximate the linear mapping as well as possible.

**Corollary 8.4.** *Theorem 1 and 2 can apply on any activations with local linearity and differentiability.*

Based on Corollary 5.1 and Lemma 5.6, it is natural to obtain the process of finding $\vec{\theta}$: by gradient descent or the closed form of Corollary 5.1. We can compute the optimal weights for the bottom Sigmiod block. On the other hand, after random initializing, the parameters of un-trained components in the Relu or Tanh blocks are assigned. This implies they can be treated as the all pre-trained case in Theorem 1 or 2. In fact, given the outputs from bottom level block then Corollary 5.1 provides weights improving the accuracy. Then it goes to the next up level block until the top, which is the forwarding steps of Back-propagation LeCun et al. (1988). Hence with initialization for un-trained component, Corollary 5.1 is essentially the same as Back-propagation.

### 8.3 A MIX OF PRE-TRAINED AND UN-TRAINED COMPONENTS

Now we first consider some of $\{f_{\Theta_j}(\mathbf{x}_j)\}_{j=1}^K$ are pre-trained and some are un-trained, and then investigate the hierarchical combination of both kinds of components. In particularly, Eq. (1) can be re-written as $g(\mathbf{x}) = w_0 \cdot \sigma(\theta_1 f_1 + \theta_2 f_{\Theta_2}) + b_0$, where $f_1$ is a pre-trained component and $f_{\Theta_2}$ is un-trained. Since $\Theta_2$ is not fixed, it can not be checked that LIC and NPC assumptions are satisfied. On the other hand, after initialization, $f_{\Theta_2}$ can be seen as a pre-trained component at any a snapshot during training phase.

**Theorem 5.** *In the end of an weight updating iteration, if the components $f_1$ and $f_{\Theta_2}$ satisfy LIC and NPC assumptions, then with high probability $\vec{\mathbf{w}}$ updated in the next iteration can improve the loss.*

*Proof.* Recall the training algorithm is the backpropagation algorithm. Also note that according to Eq. (1), the order of updating is $\vec{\theta}$ first and then $\Theta_2$. We denote in the end of iteration $i$ the value of $\vec{\theta}$ and $\Theta_2$ as $\vec{\theta}^{(iter=i)}$ and $\Theta_2^{(iter=i)}$, respectively. With randomized initialization, $\Theta_2$ is assigned as $\Theta_2^{(iter=0)}$ before the execution of the iteration 1. Then in each iteration $i \geq 1$, $g(\mathbf{x})$ is a combination of fixed parameter components. Hence this can reduce to the all pre-trained cases, and can apply Theorem 1 and 2. $\qquad\square$

**Lemma 8.1.** *For a given data set $X$, let $\vec{g} := (g_1, ..., g_N)$ and $\vec{y} = (y_1, ..., y_N)$. If $\langle \vec{g}, \vec{g} - \vec{y} \rangle \neq 0$, then there exists $\alpha \in \mathbb{R}$ s.t.*

$$\sum_{i \in [N]} (\alpha g(x_i) - y_i)^2 < \sum_{i \in [N]} (g(x_i) - y_i)^2$$

*Proof.* It is equivalent to show the inequality

$$\sum_{i \in [N]} (\alpha g(x_i) - y_i)^2 - \sum_{i \in [N]} (g(x_i) - y_i)^2 < 0$$

has a real number solution.

$$\sum_{i \in [N]} \left[ (\alpha g(x_i) - y_i)^2 - (g(x_i) - y_i)^2 \right]$$

$$= \left( \sum_{i \in [N]} g(x_i)^2 \right) \alpha^2 + \left( -2 \sum_{i \in [N]} g(x_i) y_i \right) \alpha + \left( - \sum_{i \in [N]} g(x_i)^2 + 2g(x_i) y_i \right)$$

$$= \langle \vec{g}, \vec{g} \rangle \alpha^2 + (-2\langle \vec{g}, \vec{y} \rangle) \alpha + (-\langle \vec{g}, \vec{g} \rangle + 2\langle \vec{g}, \vec{y} \rangle).$$

This is a quadratic inequality of $\alpha$, hence if

$$(-2\langle \vec{g}, \vec{y} \rangle)^2 - 4 (\langle \vec{g}, \vec{g} \rangle) (-\langle \vec{g}, \vec{g} \rangle + 2\langle \vec{g}, \vec{y} \rangle) \geq 0,$$

then there exists at least one real solution. $\qquad\square$

Now we first consider some of $\{f_{\Theta_j}(\mathbf{x}_j)\}_{j=1}^K$ are pre-trained and some are un-trained, and then investigate the hierarchical combination of both kinds of components. In particularly, Eq. (1) can be re-written as $g(\mathbf{x}) = w_0 \cdot \sigma (\theta_1 f_1 + \theta_2 f_{\Theta_2}) + b_0$, where $f_1$ is a pre-trained component and $f_{\Theta_2}$ is un-trained. Since $\Theta_2$ is not fixed, it can not be checked that LIC and NPC assumptions are satisfied. On the other hand, after initialization, $f_{\Theta_2}$ can be seen as a pre-trained component at any a snapshot during training phase.

## 8.4 GENERALIZATION ERROR ANALYSIS

**Theorem 4.** *Assume pre-trained components $\{f_j\}_{j=1}^K$ satisfy LIC and NPC. Let $\{GE(f_j)\}_{j=1}^K$ be corresponding generalization errors of $\{f_j\}_{j=1}^K$, and $\Theta_{(L)} \circ \sigma_{(L)} \circ \cdots \circ \sigma_{(1)} \circ \Theta_{(0)}(f_1, ..., f_K)$ be the composite neural network. Denote the generalization error, $\mathbf{E}\{L(\Theta_{(L)} \circ \sigma_{(L)} \circ \cdots \circ \sigma_{(1)} \circ \Theta_{(0)}(f_1, ..., f_K))\}$, of the composite neural network as $\mathbf{E}\{L_{\Theta, f_1, ..., f_K}\}$. Suppose the learned weights obey the normal distribution. Then with high probability, there exist a setting of $\{\Theta_{(L)}^*, ..., \Theta_{(0)}^*\}$ such that $\mathbf{E}\{L_{\Theta, f_1, ..., f_K}\} \leq \Theta_{(L)}^*(GE(f_1), ...G_E(f_K))$.*

*Proof.* (of Theorem 4) (**Proof Sketch**) We apply the idea similar to Kawaguchi (2016): the exception of non-liner activations is same with the exception of liner activations. Previous theorems provide that with high probability there exists the solution of $\Theta_{(i)}, \forall i \in [L]^+$ s.t. each $\Theta_{(i)+1}\sigma\Theta_{(i)}$ approximates a degree one polynomial $A_{\Theta_{(i)+1}\sigma\Theta_{(i)},1}$ as well as possible. If the weights are obey the normal distribution, then $\mathbf{E}\{L_{\Theta, f_1, ..., f_K}\} \leq \Theta_{(L)}^*(GE(f_1), ...G_E(f_K))$. $\qquad\square$

