# OpenReview forum: "An Analysis of Composite Neural Network Performance from Function Composition Perspective"
_ICLR.cc/2019/Conference_

### Official Review · AnonReviewer2 · 2018-10-30
**Not ready for publication**

**Rating:** 3
**Confidence:** 4

**Review:**

The paper aims at justifying the performance gain that is acquired by the use of "composite" neural networks (e.g., composed of a pre-trained neural network and additional layers that will be trained for the new task).

I found the paper lacking in terms of writing and in terms of clarity in expressing scientific/mathematical ideas especially for a theory paper.

Example from the Abstract:

"The advantages of adopting such a pre-trained model in a composite neural network are two folds. One is to benefit from other’s intelligence and diligence, and the other is saving the efforts in data preparation and resources
and time in training"

The main results of the paper (Theorem 1,2,3) are of the following nature: if you use more features (i.e., "components") in the input of a network then you have "more information", and this cannot be bad. Here are the corresponding claims in the Abstract:

"we prove that a composite neural network, with high probability, performs better than any of its pre-trained components under certain assumptions."

"if an extra pre-trained component is added to a composite network, with high probability the overall performance will be improved."

However, this argument seems to be just about expressiveness; adding more features can be statistically problematic.

Furthermore, why is it specific to pre-trained components? Essentially the theorems are about adding any features.

Finally, the assumption that the pre-trained components are linearly independent is invalid and the makes the whole analysis somewhat simplistic.


The motivating Example 1 just shows that the convex hull of a class of hypotheses can include more hypotheses than the class itself. I don't see any connection between this and the use of pre-training.

Other examples unclear statements from the intro:

"One of distinctive features of the complicated applications is their applicable data sources are boundless. Consequently, their solutions need frequent revisions."

"Although neural networks can approximate arbitrary functions as close as possible (Hornik, 1991), the major reason for not existing such competent neural networks for those complicated applications is their problems are hardly fully understood and their applicable data sources cannot be identified all at once."

There are many typos in the paper including this one about X for the XOR function:
"Assume there is a set of locations indexed as X = {(0; 0); (0; 1); (1; 0); (1; 0)} with the corresponding values Y = (0; 1; 1; 0). Obviously, the observed function is the XOR"

---

> ### Author Response · Authors · 2018-11-07
> **A Reply to Reviewer2**
>
> 1. Thank you for comments. Actually, we considered one or more pre-trained neural network in the paper.
>
> 2. Please pardon our non- scientific/mathematical tone that we just tried to emphasize of arrival of pre-trained neural network.
>
> 3. Yes, in simple wording, it is the main claim of this paper. Many people intuitively think so, but so far no work solves this problem. On the other hand, according to our survey (the last second paragraph of Introduction), many empirical studies point out that pre-trained models are often harmful. That’s the motivation of this work.
>
> 4. As you mentioned, “adding more features can be statistically problematic”, while Reviewer 3 said “this is a very straight forward result … we can of course represent more objects”. The different comments shows the experts do not have consensus in the effect of adding objects/features and that was the motivation of this work to study the conditions of performance improvement.
>
> 5. Pre-trained components are useful and valuable, especially it is provided by reputable individuals or organizations, such as ResNet50 provided in Keras. We believe the pre-trained components will become popular soon.
> Furthermore, the performance of adopting pre-training is unclear in the literature.
> We quote from some papers for the evidence that the performance of adopting pre-training is unclear:
> In [1]: ” Even so, relatively little is known about the behavior of pretraining with datasets that are multiple orders of magnitude larger.”
> In [2]: “there remain open questions about the performance of pretrained distributed word representations and their interaction with weight initialization and other hyperparameters.”
>
> 6. The condition “linearly independent” is given to assure the result is theoretically sound, but as all we know that the output of several neural networks are hardly “linearly dependent”. So, the proposed theory is generally applicable to most neural networks.
>
> 7. To generate convex hull from several vectors, the weights must be positive and summing to 1. The weights in Example 1 are not satisfying these two conditions. (In particular, “w_1=3 and w_2=-1” is not the convex combination.) Besides, the since the weights in a pre-trained model are frozen, we can see it as a black box or a function. That is why we denote x_1x_2 as f_3 and so on.
>
> 8. In our pdf file, the X is shown as “X = {(0, 0), (0 1), (1, 0), (1, 0)}”. We have no idea why commas become semicolons. We apologize for all the other typos. We will correct them and also clarify the obscure statements.
>
>
> Reference:
> [1] D. Mahajan, R. Girshick, V. Ramanathan, K. He, M. Paluri, Y. Li, A. Bharambe, and L. van der Maaten. “Exploring the Limits of Weakly Supervised Pretraining,” ECCV2018.
> [2] I. Cases, M.-T. Luong, and C. Potts, “On the effective use of pretraining for natural language inference”, arXiv:1710.02076

---

### Official Review · AnonReviewer1 · 2018-11-02
**The result seems straight forward**

**Rating:** 3
**Confidence:** 3

**Review:**

This paper studies composite neural network performance from function composition perspective. In theorems 1, 2 and 3, the authors essentially prove that as the basis functions (pre trained components) increases (satisfying LIC condition), there are more vectors/objects can be represented by the basis.

To me, this is a very straight forward result. As the basis increases while the LIC condition is satisfied, we can of course represent more objects (the new component is one of them). I don't see any novelties here. The result is straightforward, and this should be a clear rejection.

---

> ### Author Response · Authors · 2018-11-07
> **A Reply to Reviewer1**
>
> Thank you for comments. Most people intuitively know with more object, more representation can be obtained. But according to our survey (the last second paragraph of Introduction), many empirical studies point out that pre-trained models are on average harmful. Besides, so far no work studies this issue and that is why we wanted to give a rigorous analysis of this issue.　In this paper, we consider pre-trained neural network module with all its weights frozen, without any fine-tuning, while the composite network is trained, which will become an important issue soon.
>
> Furthermore, the performance of adopting pre-training is unclear in the literature. We quote from some papers for the evidence that the performance of adopting pre-training is unclear:
> In [1]: ” Even so, relatively little is known about the behavior of pretraining with datasets that are multiple orders of magnitude larger.”
> In [2]: “there remain open questions about the performance of pretrained distributed word representations and their interaction with weight initialization and other hyperparameters.”
>
> Reference:
> [1] D. Mahajan, R. Girshick, V. Ramanathan, K. He, M. Paluri, Y. Li, A. Bharambe, and L. van der Maaten. “Exploring the Limits of Weakly Supervised Pretraining,” ECCV2018.
> [2] I. Cases, M.-T. Luong, and C. Potts, “On the effective use of pretraining for natural language inference”, arXiv:1710.02076

---

### Official Review · AnonReviewer3 · 2018-11-06

**Rating:** 3
**Confidence:** 2

**Review:**

The paper considers the problem of building a composite network from several pre-trained networks and whether it is possible to ensure that the final output has better accuracy than any of its components.

The analysis done in the paper is that of a simple linear mixture of the outputs produced by each component and then by showing that if the output of the components are linearly independent then you can find essentially a better ensemble. This is a natural and straightforward statement with a straightforward proof. It is unclear to me what theoretical value does the analysis of the paper add. Further the linear independence assumption in the paper seems very strong to make the results of value.

Further the paper seems very hastily written with inconsistent notation throughout making the paper very hard to read. Especially the superscript and the subscript on x have been jumbled up throughout the paper. I recommend rejection and encourage the authors to first clean up notation to make it readable.

---

> ### Author Response · Authors · 2018-11-07
> **A Reply to Reviewer3**
>
> Thank you for comments.
>
> 1. In fact, the paper proposes not just “a simple linear mixture of the output”, rather, the paper also considers various activation functions, such as sigmoid and tanh. In our experiment shown in Table 2, the notation σ is the sigmoid.
> The condition “linearly independent” is given to assure the result is theoretically sound, but as all we know that the outputs of several neural networks on a large dataset are hardly “linearly dependent”. The proposed theory is generally applicable to most neural networks.
>
> 2a. Most people know intuitively that add more neural network components may enhance the performance in classification and regression result, but so far, we have not seen work directly pointing to this problem. On the other hand, according to our survey (please find it in the last second paragraph of Introduction in the paper), many empirical studies point out that pre-trained models are on average harmful. That is what we believe the contribution of this paper.
>
> 2b. We also quote from some papers for the evidence that the performance of adopting pre-training is unclear:
> In [1]: ” Even so, relatively little is known about the behavior of pretraining with datasets that are multiple orders of magnitude larger.”
> In [2]: “there remain open questions about the performance of pretrained distributed word representations and their interaction with weight initialization and other hyperparameters.”
>
> 3. Yes, indeed, we spent a tremendous time (months) in conducting the experiments on our poor server with 4 GPU cards (NVIDIA 1040), and we apologize for all the writing problems in the submission and will correct all the writing problems.
>
> Reference:
> [1] D. Mahajan, R. Girshick, V. Ramanathan, K. He, M. Paluri, Y. Li, A. Bharambe, and L. van der Maaten. “Exploring the Limits of Weakly Supervised Pretraining,” ECCV2018.
> [2] I. Cases, M.-T. Luong, and C. Potts, “On the effective use of pretraining for natural language inference”, arXiv:1710.02076

---

### Meta-Review · Area_Chair1 · 2018-12-13
**Limited contribution**

**Confidence:** 5
**Recommendation:** Reject

**Metareview:**

Dear authors,

All reviewers pointed out the fact that your result is about the expressivity of the big network rather than its accuracy, a result which is already known for the literature.

I encourage you to carefully read all reviews should you wish to resubmit this work to a future conference.